# Patient experience of a virtual reality calm room in a psychiatric inpatient care setting in Sweden: a qualitative study with inpatients

Maria Ilioudi ![ORCID],[1,2] Sara Wallström ![ORCID],[1,3,4,5] Steinn Steingrimsson,[1,6] Philip Lindner,[7] Almira Osmanovic Thunström,[6] Lilas Ali ![ORCID] [1,2,3]

For numbered affiliations see end of article.

**Correspondence to**
Dr. Steinn Steingrimsson;
steinn.steingrimsson@neuro.gu.se

## ABSTRACT

**Objective** Calm rooms have been developed and implemented in psychiatric inpatient care settings to offer patients a dedicated space for relaxation in a convenient and safe environment. Recent technology developments have enabled virtual reality (VR) equivalents of calm rooms that can be feasibly deployed in psychiatric care settings. While research has shown VR environments to be efficacious in inducing relaxation, little is known how these virtual calm rooms are perceived by patients. The aim of this study was to elucidate patient experiences of using a VR calm room in a psychiatric inpatient setting.

**Design** Qualitative interview study. Semi-structured interviews were analysed using qualitive inductive content analysis, which focuses on the interpretation of texts for making replicable and valid inferences.

**Setting** Swedish hospital psychiatric inpatient care setting with a wireless, three degrees-of-freedom VR head-mounted display running a calm room application simulating nature environment.

**Participants** 20 adult patients (12 women) with bipolar disorder (n=18) or unipolar depression (n=2).

**Results** Participants experienced the use of the VR calm room as having a positive impact on them, inducing awareness, calmness and well-being. They were thankful to be offered a non-pharmacological alternative for anxiety relief. Participants also expressed that they had some concerns about how they would react emotionally before using the VR device. However, after use, they highlighted that their overall experience was positive. They also expressed that they could see potential for further development of VR technology in psychiatric care.

**Conclusions** VR technology has the potential to solve pressing logistic issues in offering calm rooms in psychiatric inpatient care. VR calm rooms appear to be appreciated by psychiatric inpatients, who value their accessibility, convenience and variety of modalities offered. Participants perceived an increase in their well-being after use.

## INTRODUCTION

Patients admitted to psychiatric inpatient wards often experience episodes of heightened emotions, which left uncared for may ultimately necessitate invasive methods

## STRENGTHS AND LIMITATIONS OF THIS STUDY

⇒ This study was conducted in an inpatient psychiatric setting with participants who were introduced to a virtual reality (VR) calm room as a new method for relaxation.
⇒ The interview data in this study provided varied descriptions of patient experiences of using the VR calm room.
⇒ There were some difficulties in engaging patients in longer interviews, which was not unexpected given the acute phase that patients were in at that time.
⇒ The short duration of engagement by the informants during interviews resulted in more inclusions (n=20) to have enough data to draw necessary conclusions.
⇒ VR calm room has the potential to be a cost-effective, non-pharmacological form of anxiety relief in inpatient psychiatric setting.

such as forced medication, restraints and one-on-one monitoring, in order to keep them from harming themselves. So-called calm rooms (sensory rooms) offer a seemingly attractive alternative to these invasive methods, and consist of multisensory environments that stimulate sensation, olfactory, auditory, vision and, can be decorated with elements such as lights, pictures, iPad, aromatic oils, blankets.[1] The purpose is to offer patients a dedicated space for relaxation, in order to induce well-being through calming exercises and distraction from distress. Guided relaxation and other instructed exercises may also be offered.[2–5] Calm rooms can also be used to provide self-help through a convenient and safe environment and, at the same time, empower the therapeutic relationship between the patient and healthcare workers at inpatient units.[1 6] Previous research indicates that calm rooms can be used for clinical purposes as a beneficial method for patients in reducing stress and inducing well-being.[1 4 5 7 8]

Despite well-known benefits, calm rooms are far from ubiquitous in psychiatric inpatient wards, presumably due to the logistic required: there needs to be a dedicated room (with an overall low occupation rate), fitted with special décor and equipment, and special usage procedures are required. Recently, research has begun exploring whether virtual reality (VR) technology can be used to create virtual calm rooms. VR refers computer-generated, perceived three-dimensional and gaze-interactive environments, typically through the use of a head-mounted display.[9] VR technology is increasingly being used in the treatment of mental disorders and numerous interventions have been developed and tested.[10–14] The term 'virtual reality' has been used to describe a VR environments may promote relaxation through visualisation, engagement and immersion into pleasant virtual environments (often nature simulations), and by removing users from stressful situations.[15]

There are some clinical studies involving VR calm rooms in psychiatric settings, which show the efficacy of VR environments for mental health conditions.[8 16–18] Moreover, studies have also shown the efficacy[19–21] of VR in treating mental conditions, including social anxiety disorder,[22 23] post-traumatic stress disorder,[24] generalised anxiety disorder (GAD)[8] and phobias.[12 25] VR technology has also been used to provide immersive experiences for relaxation.[26–29] A previous study has suggested that VR nature relaxation immediately reduced negative affective states and psychological stress in patients with psychiatric disorders.[18] In addition, Repetto et al[8] showed, in a VR pilot study for the treatment of GAD, that using VR may result in better clinical outcomes at the end of the treatment. The participants also answered that they were very satisfied to use the device in every-day life's situations to relax.[8]

VR technology solves logistic issues since it can be used anywhere[17 20] and is affordable in comparison to refurbishing a room at a psychiatric inpatient care setting.[12] The technology has also been shown to be convenient and safe in psychiatric inpatient care settings since it is made of harmless materials and is usually used with supervision until the patient is trusted on his or her own.[30] VR relaxation settings provide a sense of presence, pleasure, activation and engagement and a personalised experience.[31] However, the appropriateness of VR interventions in acute conditions has not been studied extensively. In addition, the healthcare professionals need training and involving the multidisciplinary team in decision-making for VR implementation in the ward. Both patients and staff expressed a practical issue about the access to private space to provide the VR application[32]

There is a lack of knowledge of how users in psychiatric inpatients perceive the use of VR-based relaxation technologies. Therefore, the aim of this study was to elucidate patient experience of using a VR calm room in psychiatric inpatient care.

## MATERIAL AND METHODS

### Study design
This study had a qualitative and descriptive design. Semi-structured interviews were conducted and analysed using a qualitive content analysis based on inductive theory. This study was part of a larger research project aimed at evaluating the effect of VR and physical calm rooms in psychiatric inpatient care.[2]

### Setting and participants
Inclusion criteria were age ≥18 years, admitted for psychiatric inpatient care at a care setting specialised in bipolar disorder in the western region of Sweden (Sahlgrenska University Hospital, Gothenburg) and having previously used the VR calm room at least one time. Due to a lack of empty beds in other wards, patients with other psychiatric diagnoses were also included in this study. A total of 25 patients fulfilled the inclusion criteria and were asked to participate in the study by the first author. Of those, five declined and 20 agreed to an interview. There was no overlap in samples between the current study and a previously reported trial.[2]

### Procedure
The VR calm room was available at all times on the ward for the patients to use after consulting with the staff. After the first use, the patients were asked to participate in the study. Patients who used the VR calm room received instructions on the VR headset's functions and how to navigate in the virtual environment. To engage with the device, patients used a wireless, three degrees-of-freedom VR head-mounted display (an Oculus Go) running the calm room application developed by Mimerse. Example screenshots of the environment are shown in figure 1. Patient representatives were involved in the design process of the VR environment. The application offers a selection of different soothing nature environments (eg, a desert, a forest beach) that can be tailored through, for example, the day and night cycle and dynamic weather. In addition, users may engage in breathing exercises, mindfulness programmes and relaxing music. The user can interact with the environment by looking at different objects inside the programme and using the handheld controller to adjust preferences and build individualised scenery. The interactive smorgasbord approach was chosen in light of research showing that users have different preferences for different virtual nature environments,[33] as well as non-VR research showing the importance of matching soothing music to user preferences.[34]

### Patient and public involvement
No patient involved.

### Data collection
The individual interviews all followed a semi-structured interview-guide developed by the authors. The interviews were conducted between October 2021 and May 2022 by the first author (MI). Interviews took place at the psychiatric inpatient care setting during patients' care time. The

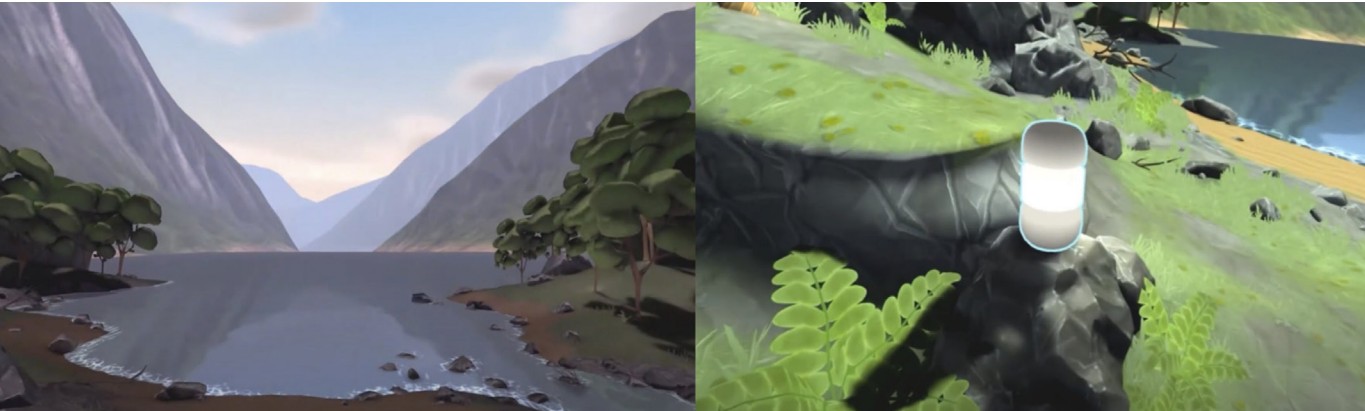

**Figure 1** Screenshots from the virtual reality calm room.

mean duration of the interviews was 15.15 (range, 10–24) min. The interviews were recorded and transcribed verbatim. The participants were asked, for example, how they experienced the calm room, was there anything that worried them when using the VR calm room, how do they think an alternative to the VR calm room can be developed and what kind of difference do they think using the VR calm room has made for their situation. After 20 interviews, data saturation was attained.

## Data analysis

Data were analysed using qualitative inductive content analysis, which search for patterns in the text, aiming to acquire a theoretical understanding from collected interview data.[35] The inductive approach entails that text is analysed based on the participants' narratives through grouping meaning units containing aspects related to each other in content and context.[36] Baxter[37] defines 'meaning unit' as the constellation of words or statements that relate to the same central meaning. A condensed description of the phenomenon is achieved by emphasising differences and similarities within the data. A theme is a thread of an underlying meaning through meaning units or codes on an interpretive level. Meaning-units were identified according to the study aim, which were then condensed and labelled with codes.[37]

First, the first and last authors (MI and LA) familiarised themselves with the material by reading the transcriptions several times in order to grasp the central meaning of the texts. Through careful reading, relevant sentences and phrases to the study aim were identified. These were then used in the constellation of meaning units and to begin the coding process. The codes were then sorted and abstracted into themes and subthemes. To ensure trustworthiness of the themes, the first and last author worked closely with the analysis. The other authors validated the text and discussed the themes and subthemes until consensus was reached.

## Ethical considerations

All participants received verbal and written information about the study and that they could end the interview at any time without specifying a reason. All data were treated confidentially. All ethical decisions were guided by the Declaration of Helsinki[38] and its subsequent amendments. The participants admitted to the psychiatric inpatient care setting may have been under compulsory psychiatry care. The authors were aware that the participants' mental health condition might be affected and were prepared to interrupt the research process or offer suitable support services on participant request or if this was deemed necessary.

## RESULTS

A total of 20 patients (12 women and 8 men) participated in the study. Participant characteristics are summarised in table 1.

The first overall impression of the analysis revealed that using the VR calm room had a positive impact on the participants during their period of care at the inpatient psychiatric unit. The experience raised feelings of awareness, calmness and well-being. Participants also had suggestions on further development of the VR technology in mental healthcare in the future. An overview of the themes and subthemes is presented in table 2.

**Table 1** Patient demographic and clinical characteristics (n=20)

| Variable | No. of patients |
|---|---|
| Sex | |
| Men | 8 |
| Women | 12 |
| Age, years | |
| 20–29 | 4 |
| 30–49 | 5 |
| 50–59 | 5 |
| 60–69 | 6 |
| Diagnosis | |
| Bipolar disorder | 18 |
| Unipolar depression | 2 |

**Table 2** Themes and subthemes from interviews following VR calm room use

| Feelings before and during use | Negative point of view | Potential for future development |
|---|---|---|
| Being offered an alternative to medication | Practical issues and side effects | Willing to continue use |
| Feelings of relief and relaxation | Unpleasant experiences | Recommending VR |
| Sense of independency | | |
| VR, virtual reality . | | |

## Feelings before and during use

Participants expressed that they had a positive attitude about using the calm room before the intervention and thought it would be an enjoyable part in the overall time of care at the psychiatric inpatient unit. Several participants mentioned that they preferred non-pharmacological alternatives such as VR. Furthermore, they experienced a sense of independence and gave positive feedback about the opportunity to learn new relaxation methods.

## Being offered an alternative to medication

During inpatient psychiatric care, the participants sometimes encountered boundaries in being able to achieve a peaceful state of mind on their own due to the care environment being hectic or perceived as stressful. They perceived there to be few non-pharmacological alternatives available, or any support offered to them to enhance their well-being and reduce anxiety. They therefore expressed gratitude over being offered the use the VR device as an alternative to drugs.

The participants also mentioned that use of the virtual calm room was very anxiety-relieving and therefore had a positive impact on their well-being during the time they were admitted to the inpatient psychiatry unit. Being admitted to psychiatric inpatient care was typically associated with high levels of anxiety and feelings of seclusion. The participants highlighted that inpatient psychiatric care needs alternative methods that they can be offered for anxiety reduction and self-support. The participants also stated that they were upset about the limited options as well as the traditional relaxation methods previously available and therefore really appreciated the time that they spent in the VR calm room. The VR calm room enriched their day at the psychiatric inpatient care setting by giving them a new dimension to rely on when needed. The participants described the VR rooms as a strategy with which they could find respite and develop emotional self-care in a stressful environment at the psychiatric inpatient care setting. Participants also highlighted that the VR represented something innovative in psychiatric care which gives hope for new treatment options.

…one thing is that we have a lot of time here, it's slow, you have nothing to do, you maybe feel bad, if it's

anxiety. That you get a tool and you sit for a certain time. I think that can help, because sometimes you might just go and go and go and do nothing, then it doesn't get better, there are no…you have to have multiple tools. This…virtual room, well, I think it's a good thing… P1

…But I had allot of conversations before, frequently, but it wasn't enough. But it is this with less stress and conversations. It is what I mostly believe in. But…and now this, it is a fantastic point of direction, it's what it is. Really… P3

## Feelings of relief and relaxation

Participants had varying experiences of how VR could potentially influence their mental condition in a better way. The use was described as being helpful for their condition and that it gave them a sense of a holistic approach to their feelings. They expressed feelings of joy and that it was something beautiful to experience. Participants also stated that using the VR calm room fulfilled their day because they felt that they had done something creative and worthwhile for their condition. They described the environment they experienced in the VR device as dynamic and that they found some kind of a new dimension to strategically enhance their well-being. Participants also experienced the VR environment to induce a state of calm and creating an awareness of their emotions at the time of use. They found it comfortable to go into another reality and leave the physical one behind for a while. Participants in this study usually experienced a stressful reality on a daily basis which contributed to their feelings of anxiety.

…it felt very nice to enter another world and then that I felt calmer afterwards… P1

…I thought I was on an island, anyways, and like, I enjoyed that there were mountain peaks, it was nature, it was leaves and it was sand and sea, or water. And it was light and it was dark and it was like morning, day dusk, night, sky. Yes, everything one needs… P13

## Sense of independency

The VR device was viewed as an opportunity for participants to learn new relaxation methods, such as relaxing the body, practising breathing and maintaining positive thinking. Participants expressed that it was easy to learn and practice guided meditation using the VR application. Furthermore, they mentioned that VR was easier to handle compared with usual relaxation methods that they would use in the 'real world' . Participants wished to stay longer in the VR calm room because they described their use as a new way to reach a feeling of calmness. Participants also expressed that using the VR room gave them a sense of independence in the psychiatric inpatient care setting by allowing them to make their own decisions and becoming more involved in their own care.

…It's not easy to describe. Now I'll see, you first came…completely shocked by that nature and then

feelings just grow, it's nice inside you that fills up the soul… P7

Participants indicated that the device was comfortable and flexible to use. They could, on their own, decide how they wanted their calm room to be and the duration of the session. This was all handled by themselves, which gave them a sense of independence in the psychiatric inpatient unit. Furthermore, participants expressed that being able to stay in the room at the psychiatric inpatient care setting solved logistic issues and could therefore be used at any place in the ward based on their own individual preferences. They also appreciated that they could pause, end and restart the session at any time.

> …If you're alone, if no staff is here, the VR glasses would have been good because then you can also lie down. I thought that was good. In this relaxation thing, they don't nag. In some relaxation videos it's, tie the hand, open the hand, tie, open. Here it was just, tie the hand and then you did the rest yourself. So that was enough information… P5

### Negative point of view
This theme describes the participants' negative experiences and feelings related to the use of the VR device. Some experienced unpleasant feelings at the start when first using the VR calm room. They were afraid and worried about some components of the VR nature environment such as the darkness and birds. Some of the participants also mentioned that the device was not very comfortable to wear on their head.

### Practical issues and side effects
Participants experienced negative issues with the VR device being uncomfortable and heavy to carry for a longer period of time on their head. Some participants mentioned that the VR device did not fit properly and would therefore slip during the time they used it. This affected the overall experience of VR device use. They also mentioned that sometimes they would become dizzy and feel nauseous while using the device. Participants also mentioned that these issues usually came at the start of the session and would become less common as they became more familiar with the VR environment. None of the participants interrupted their session due to cybersickness. Before using the VR device, they had their hesitations, some of them worried that they might feel a bit dizzy and uncomfortable. However, they managed to overcome that, after using the VR device for a while, finding it was not as frightening or strange as they thought that it would be. The participants expressed that, when they used the VR device, they felt more relaxed at the end of the session than before.

> …but that it slipped and then it got on the glasses and then there was no sharpness so I had to take it up all the time… P3

> …In the beginning it was difficult to relax, I thought, in the beginning I kind of was laying and kind of tossed and turned like this … and then when some time had passed I became really relaxed and then the rain came and birds and it was… I actually forgot where I was…P17

### Unpleasant experiences
Entering the VR world was for some participants an unpleasant experience because they were sometimes stressed and worried for no reason that they can recall. They were also expecting their own imagination to kick in and that is why they did not want to be left alone in the VR calm room. They indicated that they were waiting for more life and interaction in the virtual world such as with animals or other creatures.

> …At first it was a bit strange and scary because when I turned my head there was like nothing but the VR environment and then I got stressed at first because I felt it was too close and I didn't recognise myself. But then it got better and then it became a bit like my environment, my own. So it turned around and got better… P5

Furthermore, the analysis indicated that some of the participants experienced VR as frightening and an unsafe environment when specific images would appear such as birds flying everywhere or something else unexpected that might happen in the VR environment making them stay alert in case something unexpected would occur. Participants also described that one scenario in the VR was too dark, which usually led to them becoming anxious of the light and therefore interpreting the experience as being uncomfortable as it would remind them of walking into death when you see the light at the end of the session.

> …I perceived the light as…a bit like death, that you walk towards the light, that it became too much…that it became too peaceful… P4

> …Then the birds, there were so many birds, it didn't feel like a threat but it felt a bit stressful, I don't like seagulls, so it got a bit like this… P15

### Potential for future development
This theme captures participants' suggestions and recommendation on future use of a VR calm room and what kind of potentials they see for further development, for example, VR as an alternative to treatment for stress reduction.

### Willing to continue use
The VR experience was described as a new experience for the participants. They stated that they would consider using the VR device again, both in a healthcare setting and by themselves for self-care at home. They wanted to have easy accessibility to the VR device every day in the psychiatric inpatient care setting and wished that there would be several VR devices available for use. Furthermore, the participants expressed that they wished to use

VR on a daily basis as a part of their treatment. Several participants wanted to have a VR device of their own and asked if they could buy one or get access to the calm room application. The main statement that was given by the participants in this study was that they wanted to use the device again in the future.

> …This was really relaxing… I wish that it was always available to us that are bipolar or has other mental problems, to be used for treatment purpose… P19

> …I want to use it again and I was still sceptical before, I was open-minded, sceptical, but I would like to use it again if I get the chance… P20

### Recommending VR

Participants mentioned that VR could reduce anxiety and induce well-being. They recommended the use of the VR device for therapeutic purposes as they could see potential for that. Despite some frightening features, the participants also described use as a positive and not harmful experience, which is why they would recommend it to others.

> …It's relaxing so it affects everyone positively, it can be positive for everyone… P7

However, there were some participants who mentioned that they thought that the VR concept might be inappropriate for patients who have vision problems and difficulties with crisis situations, such as for some patients with psychosis. It might also be inappropriate to use when feeling too tired or trapped. Participants wished for further functions to be developed in the VR device and wanted to be able to choose different environments, not just the natural calm room that was programmed. They wanted to experience more dynamics, and visions that included more animals and other creatures than what they had experienced. The participants recommended and wished to use the VR for about 15–20 min as they felt that in that time, they could experience the full effect of anxiety relief.

### DISCUSSION

The aim of this study was to elucidate experiences of using a VR calm room by patients admitted to psychiatric inpatient care. Our findings strengthen the results of previous research, which showed VR calm rooms to increase well-being and lower distress.[26 28 39] In the current study, participants expressed feelings of relaxation, which induced their well-being, when using the VR room. Implementation of VR technology in psychiatric inpatient care settings has been shown to successfully improve patient recovery and emotional self-care[40] as a VR calm room strengthens patient choices, leading to a flexible and convenient strategy for resolving distress in an acute psychiatric setting.[2] Such methods might enhance patient engagement in their care and to take on the responsibility for recovery of their severe mental condition.

The findings support that the VR calm room reduce the experience of stress and helped patients to feel better with a sense of calm. Previous studies have shown that immersion into VR environments has positive psychological and physical effects.[17 41] In addition, VR calm room design can distract patients from stressful acute psychiatric settings.[42] The design of the VR room in our study, which aimed at creating a less hospital-like experience with features supporting familiarity and flexibility, may have contributed to the patients' sense of safety, self-control and enjoyment. Exposure to VR nature images can provide emotional well-being for patients who do not have access to outdoor activities and nature.[43] Patients stay in psychiatric inpatient care for long periods of time and VR calm rooms have the potential to offer visual access to nature and its positive effects. VR technology may solve logistics issues in the psychiatric inpatient care as patients can use the VR device in their own room in the psychiatric inpatient setting without affecting safety.[10 44] In comparison to traditional relaxation methods, participants mentioned that the VR calm room was easy to access and use. VR relaxation is appreciated by users,[45] and has become affordable and more accessible, which has led to its increasingly widespread use in psychiatric care.[42]

Participants reported several negative experiences with the use of VR. Some patients expressed that the experience was unpleasant and them being worried for no reason. They also mentioned becoming dizzy and that some of the images in the VR calm room were scary. The unpleasant experiences might be due to the new environment and unfamiliar components. Our findings align with previous studies where users reported issues with the installation process, overheating, the visual presentation and negative effects such as anxiety.[10] It is possible that some of these negative experiences may be due to personal preferences, which might be avoidable by offering the user a variety of environments.[46] Patients admitted to psychiatric inpatient care experience low satisfaction with their influence on the choice of treatment and medication.[47] Non-pharmacological methods such as physical calm rooms have safety issues; therefore, healthcare workers may become risk averse and initially worried about letting patients stay on their own in calm rooms.[48] However, activities that break up the monotony in psychiatric inpatient care settings were appreciated by the patients.[49]

The participants in this study highlighted some future recommendations and suggestions about VR calm rooms such as creation of several scenarios and device options with more modalities. This is in line with previous research which suggests that the VR scenario should be expanded to include various additional experiences such as live music scenes, scenes featuring people walking around and animals.[50] Patients recommended the use of the VR device as a therapeutic method for relaxation and stress reduction both in healthcare settings and for patients to use on their own at home. However, some participants mentioned that VR might be inappropriate

for patients with vision problems and when being too tired. Also, some recommended avoiding the use of VR in crisis psychiatric conditions due to the high levels of anxiety and lower adaptation ability in these circumstances. Self-relaxation is an important feature of the care process in psychiatric inpatient care; therefore, participants stated that they wanted to use the VR room in the future as a relaxation tool. Broader use of VR calm rooms should be investigated to explore their potential for being part of usual treatment in acute psychiatric care with individualised content. Important considerations in inpatient setting are agitation and risk for violence, which have to be evaluated in each case. Presumably, a patient who is used to the technique might benefit in an agitated state but studies on the safety of such an intervention are needed.

## Study strengths and limitations

The primary strength of this study is that it was conducted in inpatient psychiatric care. It is especially important to include patients who are diagnosed with severe acute psychiatric conditions as they are often excluded from studies due to their illness. This study had a sample of participants with bipolar diagnosis and unipolar depression. The interview data in this study provided varied descriptions of patient experiences of using the VR calm room.

The qualitative approach using content analysis[35–37] made it possible to elucidate patient experiences using a VR calm room in acute psychiatric care. Trustworthiness was established through including all authors' reflection and engagement in all steps of the analyses. Regarding dependability, the same author conducted the interviews using a semi-structured interview guide. One limitation in this study was the difficulties to engage patients in the interviews for a longer period of time. The interview time ranged between 10 and 24 min (mean 15.15). This might be due to the acute phase that patients were in at the time of their interviews and the severity in their illness; however, this was compensated by the inclusion of more participants in the study to have enough data to draw necessary conclusions. Patient representatives were involved in the design process of the VR environment, but there was no patient involvement in the subsequent processes of the study.

## Conclusion

A VR calm room provides a cost-effective, modern, convenient method for relaxation in acute psychiatric care. Patients appreciated the accessibility, convenience and variety of modalities. They also thought that it increased their well-being. VR technology provides potentialities in solving logistic issues and is cost effective. Further research is needed that includes patients in psychiatric inpatient care with a broader range of mental conditions.

## Author affiliations

[1] Region Västra Götaland, Sahlgrenska University Hospital, Psychiatric Department, Goteborg, Sweden
[2] Institute of Health and Care Sciences, Sahlgrenska Academy, University of Gothenburg, Gotheborg, Sweden
[3] Centre for Person-Centred Care (GPCC), Sahlgrenska Academy, University of Gothenburg, Gothenburg, Sweden
[4] Region Västra Götaland, Sahlgrenska University Hospital, Department of Forensic Psychiatry, Gothenburg, Sweden
[5] Centre for Ethics, Law and Mental Health (CELAM), University of Gothenburg, Gothenburg, Sweden
[6] Institute of Neuroscience and Physiology, University of Gothenburg, Sahlgrenska Academy, Goteborg, Sweden
[7] Centre for Psychiatry Research, Department of Clinical Neuroscience, Karolinska Institute and Stockholm Health Care Services, Stockholm, Sweden

**Acknowledgements** We thank the participants who generously shared their experiences in this study. The authors would also like to thank the Psychiatry Affective Department, the managers and, particularly, area manager Mathias Alvidius for their support and assistance in conducting the study. We would also like to thank William Hamilton who developed the virtual reality calm room scenario. We would also like to thank Peter Todd (Tajut Ltd., Kaiapoi, New Zealand) for third-party editorial assistance in drafting of this manuscript, for which he received financial compensation.

**Contributors** SS, PL, AOT and LA were involved in the conceptualisation of the study. LA, SW and MI developed the methodology. MI curated the study data. MI, LA and SW conducted the formal analysis. MI, LA and SW undertook administration of the project. MI and LA wrote the first draft of the article. All authors were involved in the conduct of the study, review and editing of the article, and approved the submitted version. MI is guarantor of the study.

**Funding** The project was funded by Lokala FoU-rådet i Göteborg och Södra Bohuslän with 110,000 Swedish krona. SS received funding from The Söderström König Foundation (SLS-974247).

**Competing interests** PL reports having received minor consulting fees from Mimerse, the company that developed the VR application used. Mimerse is no longer an active company.

**Patient and public involvement** Patients and/or the public were involved in the design, or conduct, or reporting or dissemination plans of this research. Refer to the Methods section for further details.

**Patient consent for publication** Consent obtained directly from patient(s).

**Ethics approval** This study involves human participants. Ethical approval was granted by the Swedish Ethical Review Authority (Dnr: 318-18). Participants gave informed consent to participate in the study before taking part.

**Provenance and peer review** Not commissioned; externally peer reviewed.

**Data availability statement** Data are available upon reasonable request. Due to local regulations, we are not able to share the data freely but raw data *omitting patient identificators* can be provided from the corresponding author on reasonable request.

**ORCID iDs**
Maria Ilioudi http://orcid.org/0000-0003-0709-7768
Sara Wallström http://orcid.org/0000-0001-7579-4974
Lilas Ali http://orcid.org/0000-0001-7027-4371

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
