## [Reviewer comments · BMJ Open]

ARTICLE DETAILS

TITLE (PROVISIONAL)	Patient experience of a virtual reality calm room in a psychiatric inpatient care setting in Sweden: a qualitative study with inpatients
AUTHORS	Ilioudi, Maria; Wallström, Sara; Steingrimsson, Steinn; Lindner, Philip Editorial Board Member; Thunström, Almira Osmanovic; Ali, Lilas

VERSION 1 – REVIEW

REVIEWER	Lambe, Sinead University of Oxford
REVIEW RETURNED	27-Jul-2023

GENERAL COMMENTS	Use of virtual reality in mental health setting, particularly on inpatient wards, has the potential to greatly improve patient care. Involving patient views on the VR interventions and how best to implement them is essential in making the most of this new technology. The authors have interviewed 20 people on an inpatient ward about their experience of using a VR calm room. Some of the findings described in the results are interesting, however there are a number of issues with the manuscript that would need to be addressed. Strengths and limitations section -The strengths and limitations are a bit thin. You could consider the broader implication of the study and real world application of the findings. The limitations consider the type of study design/analysis and the limitations associated with them. Was there any PPI involvement?-In the first point the authors say that the sample as 'fully representative patients with sever psychiatric conditions'. This isn't accurate - do you mean there was a high participant rate of eligible participants?-Similarly in the third point you describe the sample as 'broad in gender and age' - as it is a study of 20 people this is over reaching. Introduction -The introduction currently doesn't set the reader up well to understand what is being studied. I would start the introduction with a definition of a calm room in the physical sense and a VR calm room. It was unclear until the method whether the VR calm room was just a space or had guided relaxation. It would be helpful to go into detail about the trial that this study is connected to for context.-Suggesting that relaxation interventions have been explored as an alternative to nature seems inaccurate.-When the authors mention previous studies of calm room intervention have been used before it would be helpful to give descriptions of the nature of the VR calm room interventions and some more detail on their outcomes.
---

	- 'VR environments provide a digital ecosystem' - unclear what this means. - The advantages of VR are underdeveloped e.g. patients can customise their relaxation space, you can embed guided relaxation in the VR program etc. etc. -Similarly the risks or challenges have not been fully outlined, just eluded to. - Could look at a paper by Brown et al.,2021 (doi:10.2196/34225) that looked at patient and staff views about using VR on inpatient wards. There are likely other quali paper that have looked at this area worth highlighting. Method - Participant characteristics should be in the results section rather than the method. -There is not enough detail on the analysis. The choice of inductive content analysis seems unusual for this type of data and I'm not confident that these results reflect this type of analysis. Can the authors give more rationale for why it was used and more detail on how the analysis was done. Possibly also giving your epistemological position. You describe that reaching reliability and validity is an important part of the coding process however this seems at odds with the idea that there are different perspectives and the reflective nature of qualitative analysis. - More information on how participants were approached would be helpful. - The demographics table is a bit thin. It would be helpful to have other details such as ethnicity, length of admission. Could you provide some information on how much participants used the Calm room - was it just once or for a period of time? Did you include people who didn't take up the interventions as they may have more negative views or hesitations. Results -Some of the content of the results is interesting such as having an alternative to medication, giving independence, and VR providing hope for new interventions. -However the coding structure doesn't quite fit. For example one theme is 'Feelings during and after' but many of the sub themes in 'negative point of view' also relate to during and after. -The results are not sufficiently supported by quotes.
--	---

REVIEWER	Baca-Garcia, Enrique University Hospital Jimenez Diaz Foundation, Psychiatry
REVIEW RETURNED	21-Aug-2023

GENERAL COMMENTS	This pilot study is very interesting and innovative. The use of virtual reality can facilitate personalised relaxation techniques with user-configured experiences. In this sense, I would like the authors to elaborate more on whether it is possible to introduce different VR scenarios, now or in the future. Another comment for the authors, which they can include in the text if they see fit. Do they think that the use of VR at the beginning of admission can reduce agitation episodes?
--

VERSION 1 – AUTHOR RESPONSE

Reviewer nr 1	Remarks
Strengths and Limitations	
The strengths and limitations are a bit thin. You could consider the broader implication of the study and real world application of the findings. The limitations consider the type of study design/analysis and the limitations associated with them. Was there any PPI involvement?	We have added a point to the section about the application of the findings, and added information about PPI to the manuscript on page 25.
In the first point the authors say that the sample as 'fully representative patients with sever psychiatric conditions'. This isn't accurate - do you mean there was a high participant rate of eligible participants?	Thank you for the comment. Sentence about the sample has been revised.
Similarly in the third point you describe the sample as 'broad in gender and age' - as it is a study of 20 people this is over reaching.	Thank you, this has now been edited.
Introduction	
The introduction currently doesn't set the reader up well to understand what is being studied. I would start the introduction with a definition of a calm room in the physical sense and a VR calm room. It was unclear until the method whether the VR calm room was just a space or had guided relaxation. It would be helpful to go into detail about the trial that this study is connected to for context.	Thank you for this comment, we have now clarified and revised the introduction.
Suggesting that relaxation interventions have been explored as an alternative to nature seems inaccurate.	This part has been deleted.
When the authors mention previous studies of calm room intervention have been used before it would be helpful to give descriptions of the nature of the VR calm room interventions and some more detail on their outcomes.	Thank you for your remark, we have now clarified this on page 8.

VR environments provide a digital ecosystem' - unclear what this means.	This part has been deleted.
The advantages of VR are underdeveloped e.g. patients can customise their relaxation space, you can embed guided relaxation in the VR program etc. etc	Thank you, this has now been edited.
Similarly the risks or challenges have not been fully outlined, just eluded to	We have added more details on the risks and challenges.
Could look at a paper by Brown et al.,2021 (doi:10.2196/34225) that looked at patient and staff views about using VR on inpatient wards. There are likely other quali paper that have looked at this area worth highlighting.	Thank you for your recommendation, this has now been added.
Method	
Participant characteristics should be in the results section rather than the method	Thank you for your comment. We have moved this to the result section as you suggested.
There is not enough detail on the analysis. The choice of inductive content analysis seems unusual for this type of data and I'm not confident that the results reflect this type of analysis. Can the authors give more rationale for why it was used and more detail on how the analysis was done. Possibly also giving your epistemological position. You describe that reaching reliability and validity is an important part of the coding process however this seems at odds with the idea that there are different perspectives and the reflective nature of qualitative analysis.	Thank you for your comment. We have now revised this section.
More information on how participants were approached would be helpful.	Thank you for this remark. Participants were admitted to the psychiatric inpatient care and asked to participate by the first author, this has been added on page 7.
The demographics table is a bit thin. It would be helpful to have	We agree that it would have been interesting to include these data. However, the ethical

other details such as ethnicity, length of admission. Could you provide some information on how much participants used the Calm room - was it just once or for a period of time? Did you include people who didn't take up the interventions as they may have more negative views or hesitations.	approval did not include retrieval of these data so we do not have the information on ethnicity. As the aim of the study was to elucidate patient experience of using a VR calm room we did not include any patients that did not use the VR calm room.
Results	
Some of the content of the results in interesting such as having an alternative to medication, giving independence, and VR providing hope for new interventions. -However the coding structure doesn't quite fir. For example one theme is 'Feelings during and after' but many of the sub themes in 'negative point of view' also relate to during and after. -The results are not sufficiently supported by quotes.	Thank you for this remark. We can understand that it was confusing because the subtheme in the negative point of view was also labelled with "feelings" we have now changed that to experience and hope it is now clearer to why this is a different subtheme and that the quotes also makes sense now
Reviewer nr 2	Remarks
This pilot study is very interesting and innovative. The use of virtual reality can facilitate personalised relaxation techniques with user-configured experiences. In this sense, I would like the authors to elaborate more on whether it is possible to introduce different VR scenarios, now or in the future. Another comment for the authors, which they can include in the text if they see fit. Do they think that the use of VR at the beginning of admission can reduce agitation episodes?	We thank the reviewer for describing the study as interesting. Agitation with risk of violence is perhaps less likely to be effective if the patient is not used to VR but we have added a comment in the discussion. The application we used has several scenarios as described in the method section and we have commented in the discussion that scenarios could be individualized which is of course important for successful further development of the method in inpatient settings.

All authors agree with the changes made and we hope that these corrections fulfil the requirements as per reviewer comments. Please feel free to contact me if there should be any questions or concerns. We are looking forward to your response.

VERSION 2 – REVIEW

REVIEWER	Lambe, Sinead University of Oxford
REVIEW RETURNED	10-Nov-2023

GENERAL COMMENTS	Thank you for making the changes based on the previous review. I have added some comments below. There are a number of typos in the manuscript. I have list some at the end but it would be worth having the paper proof read thoroughly. Introduction I would soften the second part of this sentence 'However, VR application may not suit everyone or is not appropriate to acute conditions.' Instead perhaps say it 'may not be appropriate to acute conditions'. As there is little evidence around this and some people who are acutely unwell may find the VR calm room relaxing and helpful. Also as your sample are inpatient the findings support it's use in acute settings. Method Procedure – can you just give a bit more detail on what participants had to do. Was the VR available on the ward and you approached people that had used it or were they consented to the study and then given the opportunity to try the VR calm room for a set amount of time and feedback? Did they use it once/multiple times? More details on this would be helpful. The PPI involvement section is very thin. Great if participants were involved in the design of the environments. However, was there any PPI involvement in your study (e.g. the design or analysis etc)? If not I would add this as a limitation and remove the PPI section you have. You can add the line the patients were involved in the design of the VR environments to the description of the VR in the procedure. Results More quotes are needed to support your findings. These could be added to a table if you are struggling with the word limit. Discussion Page 23 – Not accurate to describe the study as being conducted 'with fully representative patients'. It is also inaccurate to say it was a broad sample based on both genders being represented. It is a small sample of a specific and important patient group, which is fine and appropriate for a qualitative study.
---

REVIEWER	Baca-Garcia, Enrique University Hospital Jimenez Diaz Foundation, Psychiatry
REVIEW RETURNED	05-Nov-2023

GENERAL COMMENTS	The authors have addressed all comments adequately.
---

VERSION 2 – AUTHOR RESPONSE

We are grateful for the many clarifying and helpful comments from Reviewer 1, allowing us to make continued revisions that have enhanced the clarity and completeness of the manuscript. Please find a complete listing of the changes made in the table below. All changes in the manuscript are highlighted in yellow. The manuscript exceeded the word count because of the qualitative design of the study and the citations to support the findings.